# Cost-effectiveness analysis of percutaneous coronary intervention for single-vessel coronary artery disease: an economic evaluation of the ORBITA trial

Victoria McCreanor [1,2] Alexandra Nowbar [3,4] Christopher Rajkumar,[3,4] Adrian G Barnett [1] Darrel Francis,[3,4] Nicholas Graves,[5] William E Boden,[6,7,8] William S Weintraub,[9,10] Rasha Al-Lamee,[3,4] William A Parsonage [1,11]

► Prepublication history and additional materials for this paper are available online. To view these files, please visit the journal online (http://dx.doi.org/10.1136/bmjopen-2020-044054).

For numbered affiliations see end of article.

**Correspondence to**
Professor William A Parsonage;
w.parsonage@mac.com

## ABSTRACT

**Objective** To evaluate the cost-effectiveness of percutaneous coronary intervention (PCI) compared with placebo in patients with single-vessel coronary artery disease and angina despite anti-anginal therapy.

**Design** A cost-effectiveness analysis comparing PCI with placebo. A Markov model was used to measure incremental cost-effectiveness, in cost per quality-adjusted life-years (QALYs) gained, over 12 months. Health utility weights were estimated using responses to the EuroQol 5-level questionnaire, from the Objective Randomised Blinded Investigation with optimal medical Therapy of Angioplasty in stable angina trial and UK preference weights. Costs of procedures and follow-up consultations were derived from Healthcare Resource Group reference costs and drug costs from the National Health Service (NHS) drug tariff. Probabilistic sensitivity analysis was undertaken to test the robustness of results to parameter uncertainty. Scenario analyses were performed to test the effect on results of reduced pharmaceutical costs in patients undergoing PCI, and the effect of patients crossing over from placebo to PCI due to refractory angina within 12 months.

**Setting** Five UK NHS hospitals.

**Participants** 200 adult patients with stable angina and angiographically severe single-vessel coronary artery disease on anti-anginal therapy.

**Interventions** At recruitment, patients received 6 weeks of optimisation of medical therapy for angina after which they were randomised to PCI or a placebo procedure.

**Outcome measures** Incremental cost-effectiveness ratio (ICER) expressed as cost (in £) per QALY gained for PCI compared with placebo.

**Results** The estimated ICER is £90 218/QALY gained when using PCI compared with placebo in patients receiving medical treatment for angina due to single-vessel coronary artery disease. Results were robust under sensitivity analyses.

**Conclusions** The ICER for PCI compared with placebo, in patients with single-vessel coronary artery disease and angina on anti-anginal medication, exceeds the threshold of £30 000 used by the National Institute of Health and

## Strengths and limitations of this study

► A strength of this research is that it is the first economic evaluation of percutaneous coronary intervention in patients with stable angina, using data from a randomised, placebo-controlled trial.

► This research is designed to provide useful and relevant information for decision-makers wanting to use cost-effectiveness evidence to make resource allocation decisions.

► A limitation of this study is that it uses data from only a short time horizon and extrapolation over a longer term may not be reliable.

► The research and results relate to only a subset of patients with stable coronary artery disease, and therefore may not be generalisable to a wider patient group.

Care Excellence when undertaking health technology assessment for the NHS context.

**Trial registration:** The ORBITA study is registered with ClinicalTrials.gov, number NCT02062593.

## BACKGROUND

Despite a substantial fall in age-adjusted mortality rates, the prevalence of coronary heart disease has only decreased minimally over the last 30 years.[1] Coronary heart disease represents a major burden to the UK population with an estimated over 2 million people living with the disease and leading to approximately half a million inpatient episodes per year.[1] The cost of treating coronary heart disease in the UK is substantial. Between 1991 and 2014, prescriptions for all cardiovascular diseases increased by 78%, and although the number of coronary artery bypass operations has diminished since a peak in the 1990s, the number of percutaneous coronary

intervention (PCI) procedures has increased sevenfold over the same time.[2]

According to the National Reference Costs collection, a total of 76 973 percutaneous transluminal coronary angioplasty procedures, Healthcare Resource Groups (HRGs) EY40 and EY41, were carried out by the National Health Service (NHS) in 2017–2018 and these 55 173 were coded as standard (non-complex) procedures at a total cost of £150 347 171.[3]

Published in 2017, the landmark Objective Randomised Blinded Investigation with optimal medical Therapy of Angioplasty in stable angina (ORBITA) Study was the first trial to investigate the efficacy of PCI for symptom relief of stable angina in a double blind, placebo-controlled study. The trial randomised 200 patients with angina due to stable single-vessel coronary heart disease to PCI or a placebo procedure with a primary endpoint of exercise time at 6 weeks of follow-up. The trial, which was more than adequately powered, showed that PCI when added to optimal medical therapy had no significant effect on the primary endpoint.[4] Additionally, the study showed small, but not statistically significant, placebo-controlled differences in secondary endpoints of angina frequency and health-related quality of life. Economic evaluation remains critically important in situations where clinical effectiveness of two interventions is similar but costs differ.

The ORBITA Study remains the only blinded, randomised controlled trial of the efficacy of PCI in patients with angina and offers a unique opportunity to undertake an economic evaluation of this form of therapy.

The aim of this paper is to evaluate the cost-effectiveness of PCI compared with placebo when added to optimal medical therapy in patients with angina due to severe, single-vessel coronary artery stenosis. Investing scarce resources for therapies that are not cost-effective reduces the aggregate of health in populations, as alternatives that deliver more health for the money are displaced.

## METHODS

We conducted an economic evaluation, in the form of a cost-utility analysis, using data from the ORBITA trial, to assess the cost-effectiveness of PCI in patients with stable, single-vessel coronary disease, in the context of the NHS of England.

Cost-utility analyses use health utility as the measure of health outcome. Health utility is a generic measure of a person's overall well-being, and takes a value between 1, full health, and 0, equivalent to being dead. It is measured using validated tools such as the EuroQol five-dimension quality of life instrument,[5] and enables the calculation of quality-adjusted life-years (QALYs). QALYs are calculated by multiplying the health utility of a health state by the length of time a person experiences that health state and are therefore superior to endpoints such as acute events or life expectancy, because they account for both length and quality of life. This is particularly important

for chronic conditions, where the main treatment goal may be symptom relief.

We modelled costs and QALYs arising from the treatment effects of the ORBITA Study, extrapolated to 12 months, and present the results as incremental cost-effectiveness ratios (ICERs) expressed as the cost per QALY gained.

Analyses were conducted using R statistical software (V.3.4.2) in the R Studio environment.[6 7] Economic modelling was conducted using the *heemod* package in R.[8]

### Model structure

For our analyses, we used a Markov model. Markov models include health states, which patients transition through over time.[9] Patients have probabilities of moving during each cycle. Cycle length and the total number of cycles are determined by the disease and treatment trajectory. Our model uses weekly cycles, for 52 weeks. Each health state in a model has health outcomes and costs attached, and patients accrue these as they move through the model.

For chronic diseases, Markov models have advantages over other methods such as decision trees, as they enable patients to remain in one state over multiple cycles. Decision trees, by contrast, can become unwieldy because new branches may be needed for each chance of moving between health states.[9]

For our model of treatment for stable single-vessel coronary artery disease (CAD), all patients enter the model with stable coronary disease and are treated with either medical therapy and placebo intervention, or medical therapy and PCI with stent implantation (figure 1). The model uses data from the ORBITA trial, and models extrapolated costs and health outcomes to 12 months. It enables the comparison of costs and health outcomes for these patients under different treatment scenarios.

We did not include death or myocardial infarction in the model, because previous randomised trials comparing medical therapy and PCI have shown no difference in the risk of these events in patients with stable CAD.[10 11]

We used a time frame of 1 year because, in previous open-label clinical trials comparing PCI with medical therapy for stable CAD, this is when a gain in quality of life from PCI is most pronounced. For example, in the COURAGE trial, quality of life had diverged between

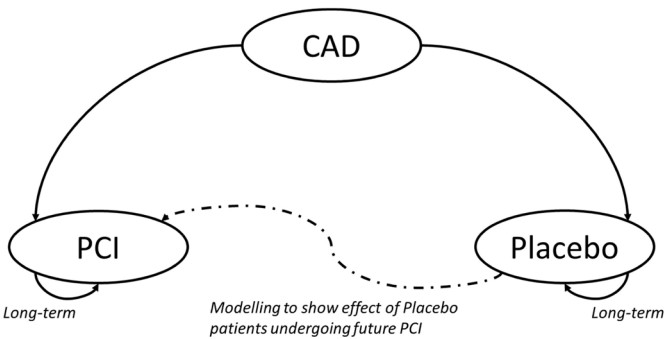

**Figure 1** Markov model of health states. CAD, coronary artery disease; PCI, percutaneous coronary intervention.

**Table 1** Health utility estimates from EQ-5D-5L data collected during the ORBITA trial; higher scores indicate better health

| Health state | Number | Utility weight | | |
| | | Mean | SE | SD |
|---|---|---|---|---|
| CAD (baseline) | 195 | 0.77 | 0.015 | 0.213 |
| Placebo | 91 | 0.81 | 0.021 | 0.221 |
| PCI | 104 | 0.83 | 0.023 | 0.212 |

Number =the number of patients in the sample, with a complete EQ-5D utility weight, used to estimate the mean, SE and SD.
CAD, coronary artery disease; ; EQ-5D-5L, EuroQol five-dimension quality of life instrument, 5-level version; ORBITA trial, Objective Randomised Blinded Investigation with optimal medical Therapy of Angioplasty in stable angina trial; PCI, percutaneous coronary intervention; SD, Standard deviation; SE, Standard error.

the randomised groups after 4 weeks with the difference sustained at 12 months before converging and becoming not clinically significant at 24 months.[12]

## Quality of life

We used the trial data for estimates of quality of life, based on all available measures of EuroQol 5-level questionnaire (EQ-5D-5L) at baseline and at completion of follow-up at 6 weeks after randomisation according to the randomised allocation (intention to treat). These are shown as health utility weights derived from the EQ-5D-5L questionnaires using the value set for the UK population, derived by Devlin *et al*[13] in the model. In the ORBITA trial, the EQ-5D-5L questionnaire was administered to patients at three time points: enrolment, pre-randomisation and follow-up. This questionnaire has been validated for use as a health utility measure, for the purposes of economic evaluation.[5] The combination of responses to the five

questions in the EQ-5D-5L is used to generate a health utility score between 0 and 1.

For the model, we used the mean health utility weight across all patients at enrolment for the CAD state and the mean health utility weight in each group, at completion of follow-up, for each of the treatment health states. We assumed that patients in these health states remained unchanged to 12 months after randomisation. The health utility weights used in the model are in table 1.

## Costs of pharmaceutical therapy

We estimated mean weekly costs of pharmaceutical therapy in both placebo and PCI groups using the trial data, and the basic price from the NHS drug tariff. The supplemental materials to the ORBITA trial paper showed the number and percentage of people in each group taking each type of medication, and the details of the medical therapy protocol (see the ORBITA trial paper in online supplemental ORBITA trial paper to online supplemental table A3 .[4] We used those figures and the national tariffs for each drug, to calculate the mean costs in each group, as summarised in table 2. The ORBITA medical therapy protocol is summarised in the supplemental materials, alongside the basic price from the January 2019 NHS drug tariff,[14] as well as unit and weekly costs for each drug.

We used average weekly costs of medical therapy in the PCI group of £2.69 per week, and £3.62 per week in the placebo group (table 2).

For the scenario analysis where patients undergoing PCI no longer require anti-anginal medication, we used an average weekly medical therapy cost of £1.11, calculated by removing costs for anti-anginal medications in the PCI group, in table 2.

**Table 2** Pharmaceutical costs for placebo and percutaneous coronary intervention (PCI) groups using data from the ORBITA trial

| Drug | ORBITA protocol dose | PCI | | Placebo | |
| | | Proportion taking | Mean weekly cost | Proportion taking | Mean weekly cost |
|---|---|---|---|---|---|
| Aspirin | 75 mg OD | 0.99 | £0.13 | 0.97 | £0.13 |
| Atorvastatin | ≥40 mg OD | 0.97 | £0.23 | 0.96 | £0.23 |
| Clopidogrel* | 75 mg OD | 1.00 | £0.33 | 0.98 | £0.32 |
| Perindopril* (if known hypertension) | ≥4 mg OD | 0.81 | £0.43 | 0.79 | £0.42 |
| Bisoprolol* | ≥5 mg OD | 0.81 | £0.12 | 0.76 | £0.11 |
| Amlodipine* | ≥5 mg OD | 0.91 | £0.15 | 0.89 | £0.14 |
| Isosorbide mononitrate slow-release* | 25 mg OD | 0.66 | £0.16 | 0.66 | £0.16 |
| Nicorandil | 10 mg BD | 0.48 | £0.38 | 0.59 | £0.47 |
| Ranolazine | 500 mg BD | 0.07 | £0.76 | 0.14 | £1.63 |
| Mean weekly total cost | | | £2.69 | | £3.62 |

*= or equivalent.
BD, twice daily; OD, once daily; ORBITA trial, Objective Randomised Blinded Investigation with optimal medical Therapy of Angioplasty in stable angina trial.

## Cost of PCI

The national tariff sets out the prices and payment rules used by NHS providers and commissioners of care, to deliver cost-effective care.[15] We used the 2019/2020 HRG reference costs for standard percutaneous transluminal coronary angioplasty (code EY41D), £1782.[16 17] That group includes PCI, with insertion of one or two drug-eluting stents, in patients with up to three comorbidities, such as diabetes and hypertension.

## Cost of cardiology clinic visits

We included costs of the ongoing visits to cardiology clinics. We estimated that those undergoing PCI would attend once, 3 months after their procedure, and those who were treated with placebo would attend at 3, 6 and 9 months. These visits were costed according to the 2019/2020 NHS National Tariff for outpatient cardiology attendances by a single professional at £78 per patient per visit.[16 17]

## Model outcomes

When the Markov model is run, the costs and health outcomes arising from the patient's transition through the health states are summed to estimate the total costs and health outcomes for each treatment: optimal medical therapy plus PCI or optimal medical therapy plus placebo. Results are presented as an ICER. ICERs are simple ratios dividing the change in costs and the change in health outcomes resulting from an investment in a new service or health technology, in this case the use of PCI. ICERs show the additional costs required to achieve one additional unit of health benefit, one QALY, and are expressed as the cost per QALY gained.

ICERs are assessed against a threshold for cost-effectiveness. In the UK, the National Institute for Health and Care Excellence (NICE) currently uses a threshold of £20 000–£30 000 per QALY gained, with an accepted upper limit of £30 000, which we used for our analyses.[18]

It is usual to discount future costs and health outcomes when running health economic models.[19] Future costs and health outcomes were discounted at a rate of 3.5% per annum, as recommended by NICE.[20]

## Probabilistic sensitivity analysis

To test the sensitivity of the model to uncertainty in parameter estimates, we conducted probabilistic sensitivity analyses.[9] For each parameter where there was uncertainty in the mean, we created a distribution around our baseline estimate.

We modelled uncertainty in the costs of pharmaceuticals by varying the percentages of people prescribed each drug type, using beta distributions. We did not vary the dosages as we felt this might create unrealistic combinations of prescriptions and dosages which would not reflect reality. Conventionally, gamma distributions are used for healthcare costs, due to their skewed shape which allows for a small number of patients to incur very high costs. However, we did not feel that a gamma distribution would

be appropriate for pharmaceutical costs, because there is likely to be little variation across patients. NHS costs for pharmaceuticals are low and most people follow similar treatment regimes.

We used a normal distribution to model uncertainty in the estimates of health utility, using the mean and SE from the EQ-5D-5L questionnaire (see table 1).

For the costs of the PCI procedure, we used only the lowest bracket of HRG EY41. This relates to the least complex patients with the fewest comorbidities. The ORBITA patients were not complex, due to design and inclusion criteria of the study, so we did not consider it appropriate to model higher procedure costs in this analysis. Similarly, we did not model any uncertainty in the cost of cardiology outpatient visits.

We took 5000 random samples from the distributions for each relevant parameter, generating 5000 ICERs. We calculated the probability of cost-effectiveness by calculating the proportion of the simulated ICERs that fall below the cost-effectiveness threshold of £30 000 per QALY gained.

## Scenario analyses
### PCI for refractory angina in control patients

We tested the effect on the economic outcome of a scenario of patients in the control group returning with refractory symptoms requiring PCI. We modelled increasing proportions of control patients returning for PCI within 12 months in increments of 20% and recalculated the ICER for comparison with the base-case analysis that assumed no crossover.

### Reduced pharmaceutical cost of treating angina in patients following PCI

We also tested the effect on the economic outcome of a scenario where patients treated with PCI would require less anti-anginal therapy than control patients. We repeated the base-case model removing all costs for anti-anginal drugs (but not costs of antiplatelet and lipid-lowering drugs) in the patients undergoing PCI from the time of the procedure until 12 months.

## Patient and public involvement
### How was the development of the research question and outcome measures informed by patients' priorities, experience and preferences?

This paper is a secondary analysis of the ORBITA Study. The original ORBITA Study was designed in close cooperation with patients and the public. The patient and public coordinators at the National Institute for Health Research Imperial Biomedical Research Unit and the Research Design Service were engaged in reviewing the trial protocol and patient information documents including the patient information leaflet and consent form.

### How did you involve patients in the design of this study?

The ORBITA focus group, composed of patients who participated in ORBITA, have provided input and support with secondary analyses.

**Table 3** Cost-effectiveness results for a cohort of 1000 patients

| Treatment | Total costs | Total QALYs | Cost difference | QALY difference | ICER |
|---|---|---|---|---|---|
| Placebo | £410 405 | 796.092 | | | |
| PCI | £1 995 418 | 813.661 | £1 585 012 | 17.569 | £90 218* |

*ICER calculated prior to rounding.
ICER, incremental cost-effectiveness ratio; PCI, percutaneous coronary intervention; QALYs, quality-adjusted life-years.

### Were patients involved in the recruitment to and conduct of the study?

No. There was public and patient involvement in aspects of planning the ORBITA trial, however patients were not involved in recruitment or running of the study.

### How will the results be disseminated to study participants?

Results will be fed back to the ORBITA focus group and disseminated through formal publications.

### For randomised controlled trials, was the burden of the intervention assessed by patients themselves?

The burden of the intervention was not specifically assessed by patients. However, the ORBITA trial, patient information and consent forms were designed in collaboration with patient and public coordinators, and all patients gave informed consent to participate.

## RESULTS

### Baseline model outcomes

The baseline cost-effectiveness results are in table 3. The results show an increase in costs for the PCI group compared with the placebo group, which is accompanied by only a very small health gain of 18 QALYs per 1000 patients. The estimated cost-effectiveness ratio is £90 218 per QALY gained when using PCI compared with placebo in addition to medical therapy, in this group of patients. This is far higher than the £30 000 threshold used by NICE, and therefore, in these patients, use of PCI would not be considered cost-effective.

### Scenario analyses: varying the percentage of placebo group patients returning for PCI

The results for this scenario analysis, where an increasing proportion of patients in the placebo group go on to

receive PCI within the year, are shown in table 4. The results show that the ICER remains above £30 000 per QALY gained when even 80% of patients return to undergo PCI within the first year following initiation of anti-anginal therapy.

### Scenario analysis: lower medical therapy costs following PCI

The results for the scenario analysis where those undergoing PCI are able to stop all anti-anginal medications are in table 5. In this scenario, the ICER remains high, at £85 576 per QALY gained.

### Results of the probabilistic sensitivity analysis

The results of the probabilistic sensitivity analysis are shown graphically in figure 2, and summarised in table 6.

In figure 2 there is only one point for the placebo group, because this is the comparison group. It is clear from the plot that fewer blue points fall under the £30 000 threshold for cost-effectiveness than fall above it. This indicates that PCI is unlikely to be cost-effective at that threshold, compared with the placebo in patients on anti-anginal therapy.

Table 6 confirms these results showing that PCI was cost-effective compared with placebo in only 11% of simulations. There is a low probability of PCI being cost-effective in this group of patients with single-vessel CAD.

## DISCUSSION

This study describes the cost-effectiveness of PCI compared with placebo when added to optimal medical therapy, using data derived from the only double blind, randomised trial of PCI in patients with stable single-vessel CAD.

**Table 4** Cost-effectiveness results for a cohort of 1000 patients, where the percentage of placebo patients returning for PCI within 1 year is varied

| Scenario: per cent crossing over from placebo | Total costs | Total QALYs | Cost difference* | QALY difference* | ICER† |
|---|---|---|---|---|---|
| 20 | £740 538 | 797.87 | £1 254 880 | 15.791 | £79 469 |
| 40 | £1 070 460 | 799.815 | £924 958 | 13.846 | £66 804 |
| 60 | £1 399 549 | 802.023 | £595 869 | 11.637 | £51 203 |
| 80 | £1 725 707 | 804.751 | £269 711 | 8.91 | £30 271 |

*Compared with the PCI group.
†ICER calculated prior to rounding.
ICER, incremental cost-effectiveness ratio; PCI, percutaneous coronary intervention; QALYs, quality-adjusted life-years.

**Table 5** Cost-effectiveness results for a cohort of 1000 patients, where those undergoing PCI have stopped all anti-anginal medical therapy

| Treatment | Total costs | Total QALYs | Cost difference | QALY difference | ICER |
|-----------|-------------|-------------|-----------------|-----------------|------|
| Placebo | £410 405 | 796.092 | | | |
| PCI | £1 913 852 | 813.661 | £1 503 447 | 17.569 | £85 576* |

*ICER calculated prior to rounding.
ICER, incremental cost-effectiveness ratio; PCI, percutaneous coronary intervention; QALYs, quality-adjusted life-years.

There are three important findings. First, the baseline analysis shows, with a high level of certainty, that PCI for angina relief, in patients with single-vessel coronary disease on anti-anginals, requires a cost per extra QALY that exceeds thresholds typically used for cost-effectiveness in the NHS. Second, even if PCI eliminated the need for anti-anginal therapy, this has minimal effect on cost-effectiveness. Finally, even if placebo patients were to present with symptoms requiring PCI further down the line, it would require this to happen in more than 80% of patients for the placebo arm to become less cost-effective than the PCI arm. These results appear to be driven by the relatively small difference in quality of life improvements in the PCI group, compared with placebo.

Our baseline analysis generated an ICER of £90 218 per QALY gained when comparing PCI with placebo, and this exceeds the threshold of £30 000 often used by NICE when considering the cost-effectiveness of treatments. Supporting the baseline estimate, the probabilistic analysis demonstrates a very high level of certainty in the model outcomes, with less than 12% of simulations favouring routine use of PCI in this patient group. There has been vigorous debate about this threshold recently. Research from the Centre for Health Economics at the University of York, focusing on opportunity cost, or what is foregone when investment in a new technology or service displaces current services, suggests that the threshold should be about £13 000 per QALY gained.[21–23]

This means that investments in new services with ICERs above £13 000 per QALY gained would result in overall harm to NHS patients, as resources would be drawn away from services which would generate more QALYs for the same investment.

These findings support, on a cost-effectiveness basis, the strategy of anti-anginal medication as first line, as advised by international guidelines.[24 25] In clinical practice, non-PCI patients might need additional visits to maintain anti-anginal therapy levels similar to ORBITA. However, even when notional costs of such additional visits are added, the magnitude of the difference between the ICER and the cost-effectiveness threshold suggests that the non-PCI approach remains economically advantageous.

The ORBITA Study protocol set out to continue medical therapy unchanged until completion of clinical follow-up at 6 weeks following randomisation. To account for the possibility that patients treated with PCI would require less anti-anginal therapy over a longer horizon, we repeated the analysis with a scenario where the costs of all anti-anginal drugs were withdrawn in the PCI group but continued in the placebo group. Because drugs used to treat angina are relatively cheap, this had a minimal effect on the ICER for PCI, which was reduced to £85 576 per QALY.

Another important consideration is that the relatively short 6-week clinical follow-up of the ORBITA Study may have masked longer term clinical benefits of PCI over medical therapy. One specific concern is that, over a longer horizon, patients may experience more angina symptoms than detected at 6 weeks, and that the placebo effect may attenuate over time. To allow for this in our economic evaluation, we explored the proportion of patients that would need to return requiring PCI for refractory symptoms of angina within 12 months, despite optimal medical therapy, before it would be cost-effective to provide routine PCI in all patients. We found that more than 80% of patients would need to return for PCI within 12 months, before it became

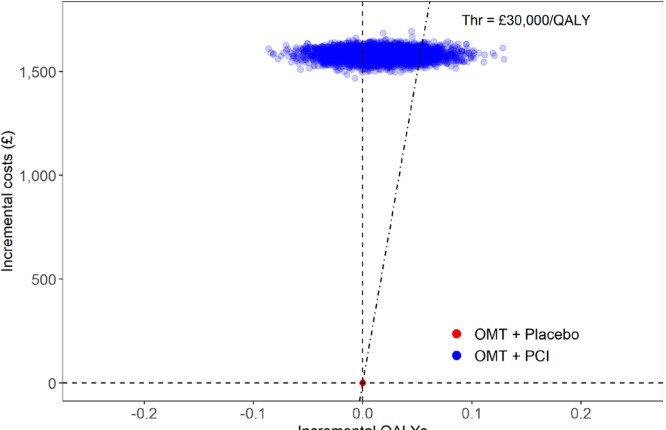

**Figure 2** Scatter plot showing results of the probabilistic sensitivity analysis for a cohort of 1000 patients. OMT, optimal medical therapy; PCI, percutaneous coronary intervention; QALYs, quality-adjusted life-years; Thr, threshold.

**Table 6** Proportion of simulations where each treatment strategy is cost-effective

| Treatment group | Cost-effective (% of simulations) |
|-----------------|-----------------------------------|
| Placebo | 89 |
| PCI | 11 |

PCI, percutaneous coronary intervention.

cost-effective to provide PCI to all patients, at the outset. This rate of crossover seems unlikely based on experience from previous randomised comparisons of PCI and medical therapy for stable CAD. For example, in the COURAGE trial, over 4.6 years of follow-up additional revascularisation occurred in 32.6% of medically treated patients compared with 21.1% of those randomised to PCI.[11] In the ORBITA trial itself, all patients had already been referred for clinical PCI, and therefore after completing their participation in the study, it was assumed that all placebo patients would then go forward for clinical PCI, and indeed most did so. This was not driven by the results of the trial because the results were not available at the time. In light of the lack of significant differences in the primary and secondary endpoints of the study related to angina, functional capacity, frequency of angina and quality of life at 6 weeks, there might be less bias towards PCI as a default treatment.[4]

## Limitations

The ORBITA trial included only patients with stable single-vessel CAD, and therefore we cannot generalise the results to patients with more complex disease. It is possible that patients with multivessel disease, more symptoms or a higher ischaemic burden may have more to gain from PCI. The ORBITA 2 trial is currently underway, and is designed to investigate the placebo-controlled efficacy of PCI in a wider clinical population.[26]

We did not model variation in costs for PCI procedures, because we felt that this best reflected the nature of the procedures and patients included in the study. However, because we used the lowest relevant HRG tariff, this would bias the model in favour of PCI. Inclusion of higher costs would have shown PCI to be less cost-effective.

We assumed that health states remained stable from the 6 weeks' follow-up to a horizon of 12 months, for the purpose of the analysis. Given that there was no difference between the groups for the key clinical endpoints in ORBITA at 6 weeks, we assumed that the effect of the intervention on quality of life was sustained over 12 months. Other factors that would affect quality of life over the longer horizon (for example other ill health) are likely to be randomly distributed between the groups and unlikely to have biased our findings.

Our model was run for a 12-month period and did not include events such as death or myocardial infarction. As noted in the methods, we did not model these events because earlier trials have demonstrated no difference, in patients with stable disease, for the two treatments examined here.[10 11]

Similarly, we did not run the model over a longer horizon, because other research has shown that improvements in symptom relief and quality of life in this patient group are most pronounced in the short term.[12] We acknowledge, however, that the outcome of the model would be sensitive to more sustained improvements in symptoms and quality of life, even if these effects are relatively small in magnitude. Publication of the outcome of studies with longer term follow-up (such as the recently published ISCHEMIA trial[27]) may help to inform models with a longer horizon. However, the open-label design of these trials leaves measures of health-related quality of life susceptible to bias that can only be controlled for in double blind, placebo-controlled studies, of which ORBITA remains the only trial of this kind at this point in time.

We were only able to partially allow for the possible effects of withdrawal of anti-anginal therapy in patients following PCI. For example, we are unable to allow for a negative effect of continuing medical therapy (medication disutility), which is likely to be greater than zero.[28] The disutility attributable to continuation of anti-anginal medication is unknown but is likely to be a complex net effect of beneficial and adverse effects. Given that patients are advised to continue with other medications (including lipid-lowering and antiplatelet agents), the effect of any disutility of continued anti-anginal therapy on our conclusions is likely to be negligible. Additionally, we have observed that in 'real-world' practice, anti-anginal drugs are often continued in patients following PCI, so our scenario analysis of withdrawal of all anti-anginal therapy in patients following PCI is also likely to be biased in favour of PCI.[29]

Our analysis is based specifically on costs relating to NHS England and cannot therefore be directly translated to other health systems. However, the costs of PCI are relatively low in the publicly funded NHS by comparison with privately funded healthcare systems. Our model is readily able to be adapted to accommodate costs incurred in different health systems.

## CONCLUSIONS

Our results show that for patients with stable single-vessel CAD and angina on medical therapy, there is a low probability that it is cost-effective to add PCI even in a healthcare system where PCI is relatively inexpensive. This conclusion is resistant to the possibility that PCI may lead to a reduction in downstream costs for anti-anginal drugs and cardiology outpatient visits and/or an increase in subsequent PCI procedures for refractory symptoms.

**Author affiliations**
[1]Australian Centre for Health Services Innovation (AusHSI) and Centre for Healthcare Transformation, Queensland University of Technology (QUT), Brisbane, Queensland, Australia
[2]Jamieson Trauma Institute, Royal Brisbane and Women's Hospital, Metro North Hospital and Health Service, Herston, Queensland, Australia
[3]National Heart and Lung Institute, Imperial College London, London, UK
[4]Imperial College Healthcare NHS Trust, London, UK
[5]Health Services and Systems Research, Duke-NUS Medical School, Singapore
[6]Boston University School of Medicine, Boston, Massachusetts, USA
[7]Harvard Medical School, Boston, Massachusetts, USA
[8]VA New England Healthcare System, Boston, Massachusetts, USA
[9]Georgetown University, Washington, District of Columbia, USA
[10]MedStar Washington Hospital Center, Washington, District of Columbia, USA
[11]Cardiology, Royal Brisbane and Women's Hospital, Herston, Queensland, Australia

**Acknowledgements** Computational resources and services used in this work were provided by the eResearch Office, Queensland University of Technology, Brisbane, Australia. The ORBITA focus group is thanked for its ongoing contribution

to the development and dissemination of secondary analyses and subsequent research following the original ORBITA Study.

**Contributors** WAP and VM planned the concept and design of the study. VM undertook the modelling and data analyses. WAP drafted the initial manuscript. All authors (AGB, AN, CR, DF, NG, RA-L, VM, WEB, WAP and WSW) reviewed the initial manuscript and provided feedback on the design and interpretation of the results. All authors (AGB, AN, CR, DF, NG, RA-L, VM, WEB, WAP and WSW) contributed to intellectual content and critical revisions of the work. All authors (AGB, AN, CR, DF, NG, RA-L, VM, WEB, WAP and WSW) give final approval of the version to be published and agree to be accountable for all aspects of the work.

**Funding** RA-L, DF, AN and CR acknowledge support from the NIHR Imperial Biomedical Research Centre (P74227).

**Competing interests** RA-L declares speakers' fees from Philips Volcano, Menarini Pharmaceuticals, outside this work. No other disclosures.

**Patient consent for publication** Not required.

**Ethics approval** This work is included under the terms of the original ethical approval for the ORBITA Study obtained from the Central London Ethics Committee (REC reference 13/LO/1340).

**Provenance and peer review** Not commissioned; externally peer reviewed.

**Data availability statement** Data are available upon reasonable request. The authors commit to making the relevant anonymised patient-level data available on reasonable request.

**ORCID iDs**
Victoria McCreanor http://orcid.org/0000-0002-0589-8521
Alexandra Nowbar http://orcid.org/0000-0001-7567-5975
Adrian G Barnett http://orcid.org/0000-0001-6339-0374
William A Parsonage http://orcid.org/0000-0002-0223-5378

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
