## [Reviewer comments · BMJ Open]

ARTICLE DETAILS

TITLE (PROVISIONAL)	Cost-effectiveness analysis of percutaneous coronary intervention for single vessel coronary artery disease: An economic evaluation of the ORBITA trial
AUTHORS	McCreanor, Victoria; Nowbar, Alexandra; Rajkumar, Christopher; Barnett, Adrian; Francis, Darrel; Graves, Nicholas; Boden, William; Weintraub, William; Al-Lamee, Rasha; Parsonage, William

VERSION 1 – REVIEW

REVIEWER	Hueb, Whady Heart Institute
REVIEW RETURNED	06-Mar-2020

GENERAL COMMENTS	To the Authors, This is an ancillary study extracted from the ORBITA Trial whose content reveals similarity in percutaneous or placebo treatments for patients with single vessel disease. The authors, after a 1-year follow-up, compared the cost effectiveness of these two therapeutic options and found no significant differences between them. Regardless of the comparisons, this group of patients is at low risk and whatever the therapeutic option, the result would not be different. Thus, a cost-effectiveness study, in the view of this reviewer, could not find any surprises. However, even if the methodology applied is adequate authors should consider some improprieties placed in this manuscript. Summary of strengths and limitations: 1- A strength of this research is that it is the first economic evaluation of PCI in patients with stable angina, using data from a randomised-controlled trial. This is not true: There are several studies that address this issue (1-3). Authors must redo this sentence 2- A limitation of this study is that it uses data from only a short time horizon and extrapolation over a longer term may not be reliable. This proposal is inconsistent. Since these are low-risk patients, long-term studies have a great chance of presenting similar results. On the other hand, since these are patients with single vessel disease, with a 5-year follow-up, researchers may find in this sample a very large percentage of multivessel disease. Thus, due to the progression of the disease, these patients are not comparable. 3- The research and results relate to only a subset of patients with stable coronary artery disease, and therefore may not be generalisable to a wider patient group. In this case, the authors should comment that these are low-risk patients with little risk of major cardiovascular events.
--

BACKGROUND:

Pag.4 of 22, line 23: The landmark study... This study is not landmark. It is a small trial, with low-risk patients and limited follow-up time, with a lot of cross-over (which makes it difficult to understand the results). Authors must remove this adjective.

Pag.4 of 22, line 33: ... only blinded.... unique opportunity to undertake an economic.... Again, it's not the only study. Regarding the blinded study, it is not credible that the operators were blinded because, during catheterizations, the operators knew that they would not do the procedure. If the patients were blinded, I think the research "touched" the ethical barrier. Therefore, I suggest that this sentence be removed.

METHODS:

Page.4 of 22, line 33..... the National Health Service of England offers health care with or without pay. Was it compared to private service? If yes make the comparison.

Page.4 of 22, line 56.... To obtain the QALYs, the SF-36, which is more complete, is often used. The use of EuroQoL leaves the study fragile. Why was this choice made?

Page.4 of 22, line 57..... QALYs are calculated by... In this case, the authors ruled out the occurrence of events and also the frequency of angina. Without these calculations, the results will have little credibility.

MODEL STRUCTURE

Page.5 of 22, line 30..... For the calculation, using the Markov model, it is necessary to apply the state of health and the occurrence of events, which include: AMI, stroke, new interventions and dead. The authors did not use this data. Why? Without this analysis, the result does not reflect the real state of Cost-effectiveness in CAD / PCI. it is necessary to include this data in the analysis.

MODEL ASSUMPTIONS:

Page.5 of 22, line 39..... We did not include death or myocardial infarction (10,11)....The authors cannot compare the studies cited as similar. These studies, (10,11), have different proposals. They aim to compare the occurrence of MACCE. Although death and infarction were similar, in these studies, an expressive number of cross-over was observed, which greatly increases the cost of treatment. In the ORBITA Trial, the cross-over was high. Thus, it is necessary to include, in this study, the calculation of events including cross-over.

Page.5 of 22, line 46..... We used a time frame of one year.... This statement is biased. Authors cannot admit a result before obtaining it. To claim that patients have gained QoL in the first year is to know the results in advance. This sentence compromises the work. It must be withdrawn.

Page.5 of 22, line 48 In the COURAGE trial quality of life... ..This phrase is unnecessary. The authors cited studied QoL and not cost-effectiveness.

Page.5 of 22, line 54 based on all available measures of EQ-5D-5L at baseline and at 6 weeks.... Os autores usaram os dados do baseline e de 6 semanas apos randomização.Para uma análise mais acurada, é necessário evolução sequencial e também no final do estudo.

Dados de seis semanas tem validade para seis semanas. É necessário inclusão dos dados do EQ-5D-5L no final do estudo.

Page.5 of 22, line 55 To our knowledge,..... This phrase is repetitive and unnecessary.

Page.5 of 22, line 60..... questionnaire was administered.....
The authors administered the questionnaire on enrolment, pre-randomization and six weeks after the procedure. This reviewer does not understand the difference of the EQ-5D-5L in the enrolment and pre-randomization. The first questionnaire is invalid. The necessary questionnaire was not carried out. The most important questionnaire is the one that should be applied at the end of the follow-up.

Page.6 of 22, line 11..... health states remained unchanged to 12 months after randomisation... .This is serious !!!! Events occurred, crossover, additional revascularization must be considered. Authors must redo the calculations.

Page.7 of 22, line 10..... Table 2 shows that the use of isosorbide is absolutely similar in both groups. Assuming that the symptoms of angina were superior to the placebo group, and the PCI corrected coronary stenosis and, consequently, ischemia, why was isosorbide used in the PCI group? It is not clear how the cost of isosorbide was removed.

Cost of cardiology clinic visits:

Page.7 of 22, line 49..... The authors report that the calculation of the follow-up price was for presumed visits. All randomized and controlled studies have clinical follow-up (quarterly or half-yearly) provided for in the protocol. this reviewer cannot consider payment for presumed visits.

Models outcomes:

Page.8 of 22, line 3.....optimal medical therapy plus PCI or optimal medical therapy plus placebo.... The authors should consider that although the placebo group did not receive Drug Eluting Stents, they were submitted to angiogram and, even, catheters were used. Price calculations cannot be virtual but real.

Page.8 of 22, line 13..... uses a threshold of £20,000 to £30,000 per QALY gained, with an.....For the purpose of comparing two strategies, the authors should consider the placebo group as a true medical group. In this study, the placebo group underwent intervention and its costs should be noted. Otherwise, this study can be considered a “virtual study” of cost effectiveness.

Probabilistic sensitivity analysis:

Page.8 of 22, line 42... We did not vary the dosages as we felt this..... This reviewer considers the ORBITA Trial a small study with 195 patients and with a very feasible follow-up. Estimating medications because of the variability of medical prescriptions does not convince this reviewer. The impression that remains is that the authors do not have this data and are making estimates of medications administered. This is another sign of work fragility.

Page.8 of 22, line 44..... For the costs of the PCI procedure.....
The authors report that they considered patients with less comorbidity and with lower health care costs. This does not make sense, because the study aims to analyze cost effectiveness between PCI versus placebo. This is not a clinical study.

Page.8 of 22, line 44..... The ORBITA patients were not complex..... Again, even though the patients were complex, this complexity would be present in both groups. This argument is not supported.

	Page.8 of 22, line 50..... We took 5,000 random samples from the distributions for each relevant parameter. This calculation does not include major cardiovascular events such as AMI, Stroke, Additional intervention and death. This must be considered. Scenario analysis: Page.8 of 22, line 58..... refractory symptoms requiring PCI..... In this case, the authors must calculate the exact number of patients. Given the small number of patients the group studied, presumed analysis of patient data is not acceptable. Page.9 of 22, line 8..... would require less anti-anginal therapy than control patients..... Again. This study must have, in the database, the exact quantity of drugs received. DISCUSSION: Page.11 of 22, line 7..... from the only double blind, randomised trial..... Here, the authors must define what is double blind. The operators were not blinded. If the placebo group operators knew that the procedure was fake then it is not double. Maybe just the patients. Page.11 of 22, line 57..... specific concern is that, over a longer horizon,.... In this case, the authors could find another population of patients with CAD. It is known that the disease is progressive and at the end of 5 years these patients could present with multivessel dysesease. Page.12 of 22, line 3..... We found that more than 80% of patients would need to return for PCI within 12 months,..... Sorry, this is creative statistics. Page.12 of 22, line 6..... in the COURAGE trial over 4.6 years.....The authors cannot consider similarity in the analysis of the two trials. Since the objectives were different and the patients were multivessel. Even so, considering a 12-month follow-up, COURAGE found a crossover occurrence close to 6% per year in the medical group and close to 4% in the PCI group. This in multivessel disease patients. Page.12 of 22, line 11..... In the ORBITA trial itself, all patients had already been referred for clinical PCI,..... Do the authors consider that all patients in the placebo group went for percutaneous treatment? If so, it invalidates the study. References 1-Cost-effectiveness of percutaneous coronary intervention versus bypass surgery from a Dutch perspective. Heart. 2015 Dec;101 (24):1980-8. 2-Cost-Effectiveness of Percutaneous Coronary Intervention in Optimally Treated Stable Coronary Patients. August 2008 Circulation Cardiovascular Quality and Outcomes 1(1):12-20 DOI: 10.1161/CIRCOUTCOMES.108.798462. 3-Comparative cost-effectiveness of surgery, angioplasty, or medical therapy in patients with multivessel coronary artery disease. Cost Effectiveness and Resource Allocation volume 16, Article number: 55 (2018)
--	---

REVIEWER	Steffan F. Stella McMaster University - Canada
REVIEW RETURNED	11-Apr-2020

GENERAL COMMENTS	Thanks for giving me the opportunity to review this manuscript.
---

	Based on clinical results from the ORBITA trial, the authors performed an economic evaluation on percutaneous coronary intervention compared to placebo for stable angina patients from the UK NHS perspective with the intent of providing evidence to inform policy makers and practice. Major comments:  - The authors implied they are using primary patient level data from ORBITA trial. The ORBITA trial reported no difference ($p=0.994$) in the EQ-5D-5L health related quality of life (QoL), with incremental in QoL of 0.03 in each group from baseline to follow up (6 weeks). The authors presented weighted utility values calculated from the trial data, however it is confusing why they aggregated the utility values for the patients in the baseline as the benefit should be assessed in each group in separate. Even correcting that, the lack of difference in effectiveness between groups persists. In this case, given the authors propose a cost effectiveness study, the denominator in the incremental cost effectiveness ratio (ICER) equation will approach to zero, which would make this type of analysis not viable or not necessary. A more pragmatic approach would take in consideration that no significant benefit in terms of QALY gained would make one to choose the less costly strategy. In the bottom-line, cost effectiveness studies are justified when one is trying to show if a given benefit in health worth the additional cost allocated on a given strategy. - Although it was previously published, the study population should be explored in the manuscript. The reader shouldn't have to refer to the original study to find out the population being studied. - The choice of a Markov model in this context is confusing. Firstly, in the manuscript, PCI and placebo are defined as health states when in fact they are decision branches in the Markov tree. Secondly, no cycle length is defined and the 12 months extrapolation in the study horizon is problematic. It is very hard to extrapolated results from 6 weeks up to 12 months. This study addresses a very specific question and population, and this assumption is completely unreliable. Moreover, the Markov model was also used to perform the probabilistic sensitive analysis, although other methods for PSA, like bootstrapping, could be more adequate given patient level information is available for the authors. - Authors need to clarify which information they want to provide. If the intent is to show cost differences between strategies, a dedicated cost study would be more appropriate. This would include a more comprehensive look into the costs related to medical therapy and PCI procedure. Ideally, averaging the costs related to each patient would provide the cost dispersion values and parameters across the sample. Additionally, one-way sensitive analysis and Tornado diagram would be of key importance to understand which are the most important cost predictors.
--	--

VERSION 1 – AUTHOR RESPONSE

Reviewer: 1

Please leave your comments for the authors below

To the Authors,

This is an ancillary study extracted from the ORBITA Trial whose content reveals similarity in percutaneous or placebo treatments for patients with single vessel disease. The authors, after a 1-year follow-up, compared the cost effectiveness of these two therapeutic options and found no significant differences between them. Regardless of the comparisons, this group of patients is at low risk and whatever the therapeutic option, the result would not be different. Thus, a cost-effectiveness study, in the view of this reviewer, could not find any surprises. However, even if the methodology applied is adequate authors should consider some improprieties placed in this manuscript.

Whilst agreeing with the reviewer that the group of patients studied in ORBITA is 'at low risk' we would dispute that the primary clinical findings of the trial were unexpected. The ORBITA study did show no statistically significant difference in several measures of clinical efficacy for medical therapy and percutaneous coronary intervention (PCI, or stenting) but in fact these findings shocked and surprised the cardiology community. This is best illustrated by the rigorous and ongoing debate that has followed the publication of the ORBITA results.

In any case the finding of clinical equipoise in no way diminishes the need for a careful and considered cost-effectiveness analysis to supplement the clinical outcome findings. Put simply, where two competing treatment options exist that lead to similar clinical outcome but at potentially different cost, there is an opportunity, and indeed a necessity, to inform decision-making for resource allocation. In this context it is crucial for both clinical and policy decision makers to be well informed regarding the incremental cost-effectiveness of these treatments and the cost implications of choosing one treatment over the other.

COMMENT

Summary of strengths and limitations:

1.1. A strength of this research is that it is the first economic evaluation of PCI in patients with stable angina, using data from a randomised-controlled trial. This is not true: There are several studies that address this issue (1-3). Authors must redo this sentence

This is a good point and we agree we should have been clearer that this is the 'first economic evaluation of PCI in patients with stable angina, using data from a randomised, **placebo-controlled** trial'. The manuscript has been amended accordingly.

1.2. A limitation of this study is that it uses data from only a short time horizon and extrapolation over a longer term may not be reliable. This proposal is inconsistent. Since these are low-risk patients, long-term studies have a great chance of presenting similar results. On the other hand, since these are patients with single vessel disease, with a 5-year follow-up, researchers may find in this sample a very large percentage of multivessel disease. Thus, due to the progression of the disease, these patients are not comparable.

We acknowledge in our discussion that extrapolation of the relatively short follow up period of the ORBITA dataset is a limitation but also justify our extrapolation to a 12-month horizon for the primary analysis based on appropriate references. The effects of medical therapy and/or PCI for angina should be fully realised within, and sustained to, 12 months. There are currently no longer term randomised, placebo-controlled studies of this problem with which to overcome this.

The reviewer may be highlighting that patients with single vessel coronary disease may progress to develop more complex disease over a longer time horizon. This is an inherent limitation of most trials of interventions for chronic diseases that may progress over time. However, given that ORBITA is a randomised trial, in which both arms received equivalent and appropriate preventive medical therapies (i.e. aspirin and lipid lowering) this does not alter the findings of our analysis as any progression of disease is likely to be equal in both study arms and accounted for in randomisation. Importantly, there is no evidence that PCI to a single specific coronary lesion influences progression of other lesions over time.

No alteration to manuscript.

1.3. The research and results relate to only a subset of patients with stable coronary artery disease, and therefore may not be generalisable to a wider patient group. In this case, the authors should comment that these are low-risk patients with little risk of major cardiovascular events.

We agree entirely with the reviewer and this is emphasised in the opening sentence of our limitation section.

No alteration to manuscript.

BACKGROUND

1.4. Pag.4 of 22, line 23: The landmark study... ..This study is not landmark. It is a small trial, with low-risk patients and limited follow-up time, with a lot of cross-over (which makes it difficult to understand the results). Authors must remove this adjective.

We respectfully suggest that what constitutes 'a landmark study' is highly subjective. The ORBITA trial has been cited over 450 times since publication in late 2017 (Google Scholar) and is rated as a 'Highly Cited Paper' (Web of Science). Web of Science metrics show that the paper has been cited in over 100 other original articles, over 50 editorials and attracted 19 letters. The paper also attracted substantial interest in public and social media with over 50 news stories including CNBC, New York Times, Time Magazine, The Conversation and Huffington Post. More importantly, the ORBITA study remains **the only** randomised, placebo-controlled trial of stenting for stable coronary artery disease. It is our view that these are reasonable grounds for considering the ORBITA trial an extremely important study in medical research.

We would be happy to use another adjective to describe the importance of the ORBITA trial but at this stage there is no alteration to the manuscript.

1.5. Pag.4 of 22, line 33: ... only blinded.... unique opportunity to undertake an economic.... Again, it's not the only study. Regarding the blinded study, it is not credible that the operators were blinded because, during catheterizations, the operators knew that they would not do the procedure.

The method of blinding of patients and clinicians in the trial, and indeed the evaluation of the efficacy of the blinding methods, are dealt with in detail in the original ORBITA trial publication.

The randomised cardiac catheterisation procedures were performed in isolation by clinical teams not involved with other aspects of the clinical trial nor the ongoing care of the patients. Patients were sedated during these procedures with additional visual and auditory masking. The efficacy of blinding was assessed following these randomised procedures in patients and their treating clinicians and again at completion of follow up before unblinding. The statistical method described by Bang et al [1] was used for these analyses and demonstrated blinding within conventionally accepted bounds (Bang index <0.20) at all time points in patients and clinicians.

Whilst double blind, placebo-controlled studies that involve procedural interventions (as opposed to pharmaceutical interventions) present obvious challenges there are, in fact, numerous examples in the medical literature of such studies being successfully undertaken.

If the patients were blinded, I think the research "touched" the ethical barrier. Therefore, I suggest that this sentence be removed.

The study protocol, including the double-blind nature of the trial, was approved by the London Central Research Ethics Committee (Ref 13/LO/1340).

No alteration to manuscript.

METHODS

1.6. Page.4 of 22, line 33..... the National Health Service of England offers health care with or without pay. Was it compared to private service? If yes make the comparison.

Our analysis was performed based on the context of the publicly funded UK National Health Service where the ORBITA trial was conducted. The method of analysis could easily be applied to different health service contexts, but this was beyond the scope of our analysis.

No alteration to manuscript.

1.7. Page.4 of 22, line 56.... To obtain the QALYs, the SF-36, which is more complete, is often used. The use of EuroQol leaves the study fragile. Why was this choice made?

The EQ-5D was chosen because it is a validated and widely used tool for estimating health utility for the purposes of health economic evaluation. The SF-36 measures quality of life, rather than health utility, and needs to be mapped to a health utility score, such as the EQ-5D, for the purposes of estimating QALYs. We acknowledge that the EQ-5D has a ceiling effect and it can be insensitive to small differences in health utility. However, in this study, the newer EQ-5D-5L was used, which will have reduced those effects.

1.8. Page.4 of 22, line 57..... QALYs are calculated by... In this case, the authors ruled out the occurrence of events and also the frequency of angina. Without these calculations, the results will have little credibility.

Angina is a symptom that is well known to affect quality of life. By 'events' we assume the reviewer is referring to acute ischaemic events such as myocardial infarction or death. It has been well established, in large randomised open-label trials, that PCI in patients with stable angina has no effect on reducing the frequency of these events. Thus, in a randomised controlled trial it would be expected that these events would be equally distributed between the randomised groups. Given that our

primary outcome is driven by *the difference* in costs between the randomised groups there is no requirement for costs associated with these events to be included in the model.

No alteration to manuscript.

MODEL STRUCTURE

1.9. Page.5 of 22, line 30..... For the calculation, using the Markov model, it is necessary to apply the state of health and the occurrence of events, which include: AMI, stroke, new interventions and dead. The authors did not use this data. Why? Without this analysis, the result does not reflect the real state of Cost-effectiveness in CAD / PCI. it is necessary to include this data in the analysis.

We respectfully disagree. See detailed response to comment 1.8 above.

MODEL ASSUMPTIONS

1.10. Page.5 of 22, line 39..... We did not include death or myocardial infarction (10,11).....The authors cannot compare the studies cited as similar. These studies, (10,11), have different proposals. They aim to compare the occurrence of MACCE. Although death and infarction were similar, in these studies, an expressive number of cross-over was observed, which greatly increases the cost of treatment. In the ORBITA Trial, the cross-over was high. Thus, it is necessary to include, in this study, the calculation of events including cross-over.

The issue of 'cross over' in the ORBITA study has been widely misunderstood. No patients 'crossed over' during the trial. Patients enrolled in the trial had been identified as those with stable angina where PCI *and* optimal medical therapy or optimal medical therapy *alone* would be a reasonable treatment options and after giving informed consent, they were then randomised to either PCI or a placebo procedure. After completion of the study and unblinding, and indeed even before the outcome of trial was known, those patients who had been randomised to a placebo procedure were offered the option of returning for PCI. A large proportion chose to do so. This does not represent 'cross over' within the clinical trial. Additionally, all measures utilised in the original study, and this economic evaluation, are acquired at completion of the trial and before patients and investigators were unblinded.

1.11. Page.5 of 22, line 46..... We used a time frame of one year.... This statement is biased. Authors cannot admit a result before obtaining it. To claim that patients have gained QoL in the first year is to know the results in advance. This sentence compromises the work. It must be withdrawn.

We are a little unclear what the reviewer means here. Whilst this is a post-hoc analysis, all data were collected prospectively.

We have altered to manuscript for clarity to include the sentence:

'We used a time frame of one year because, in previous open-label clinical trials comparing PCI to medical therapy for stable coronary artery disease, this is when a gain in quality of life from PCI is most pronounced.'

1.12. Page.5 of 22, line 48 In the COURAGE trial quality of life... ..This phrase is unnecessary. The authors cited studied QoL and not cost-effectiveness.

The health related quality of life measure used here EQ-5D-5L is used to measure health utility from which cost-effectiveness outcomes are derived. We maintain this sentence is important.

No alteration to manuscript.

1.13. Page.5 of 22, line 54 based on all available measures of EQ-5D-5L at baseline and at 6 weeks.... Os autores usaram os dados do baseline e de 6 semanas apos randomização. Para uma análise mais acurada, é necessário evolução sequencial e também no final do estudo. Dados de seis semanas tem validade para seis semanas. É necessário inclusão dos dados do EQ-5D-5L no final do estudo.

We have translated this as *'The authors used baseline and 6-week data after randomization. For a more accurate analysis, sequential evolution is necessary and also at the end of the study. Six-week data is valid for six weeks. It is necessary to include data from the EQ-5D-5L at the end of the study.'*

This was indeed what was done. We have altered the manuscript slightly for clarity and now reads

'For the model, we used the mean health utility weight across all patients at enrolment for the CAD state and the mean health utility weight in each group, at completion of follow-up, for each of the treatment health states.'

1.14. Page.5 of 22, line 55 To our knowledge,..... This phrase is repetitive and unnecessary.

This sentence has been removed

1.15. Page.5 of 22, line 60..... questionnaire was administered..... The authors administered the questionnaire on enrolment, pre-randomization and six weeks after the procedure. This reviewer does not understand the difference of the EQ-5D-5L in the enrolment and pre-randomization. The first questionnaire is invalid. The necessary questionnaire was not carried out. The most important questionnaire is the one that should be applied at the end of the follow-up.

We accept that this should have been worded with greater care, however the necessary questionnaires were indeed carried out. We could have been clearer that our model was based on the difference in health utility from enrolment to the completion of follow up. The text has been amended in two places for clarity and now reads...

'We used the trial data for estimates of quality of life, based on all available measures of EQ-5D-5L at baseline and at completion of follow up at 6 weeks after randomisation according to the randomised allocation (intention to treat).'

And...

'For the model, we used the mean health utility weight across all patients at enrolment for the CAD state and the mean health utility weight in each group, at completion of follow-up, for each of the treatment health states.'

1.16. Page.6 of 22, line 11..... health states remained unchanged to 12 months after randomisation... This is serious !!!! Events occurred, crossover, additional revascularization must be considered. Authors must redo the calculations.

See response to Comment 1.10 above. There was no 'cross over' of patients between the randomised groups for the duration of the study. Also as detailed above (response to Comment 1.8), other events will have been controlled for by randomisation.

1.17. Page.7 of 22, line 10..... Table 2 shows that the use of isosorbide is absolutely similar in both groups. Assuming that the symptoms of angina were superior to the placebo group, and the PCI corrected coronary stenosis and, consequently, ischemia, why was isosorbide used in the PCI group? It is not clear how the cost of isosorbide was removed.

This comment appears to illustrate some misunderstanding of the way the study was conducted and its eventual findings. The study randomised patients to PCI vs a placebo procedure. Further management during the double-blind phase of the study, including all medications for angina and secondary prevention, were prescribed to patients per protocol by clinicians who were blinded to the randomised treatment allocation. As detailed in Response 1.5 the blinding process was shown to have been highly effective.

The fact that prescription of antianginal medication was similar probably underlines the primary outcome of the study i.e. that PCI had no statistically significant effect on angina severity when compared to placebo.

Cost of cardiology clinic visits

1.18. Page.7 of 22, line 49..... The authors report that the calculation of the follow-up price was for presumed visits. All randomized and controlled studies have clinical follow-up (quarterly or half-yearly) provided for in the protocol. this reviewer cannot consider payment for presumed visits.

We respectfully disagree. The cost of clinical services related to the clinical trial are not relevant to a cost-effectiveness analysis. Costs related to usual care are included as described in the methods.

Models outcomes

1.19. Page.8 of 22, line 3.....optimal medical therapy plus PCI or optimal medical therapy plus placebo.... The authors should consider that although the placebo group did not receive Drug Eluting Stents, they were submitted to angiogram and, even, catheters were used. Price calculations cannot be virtual but real.

All patients were included in the study after coronary artery disease had been confirmed. Thus, the costs of angiography *within the study* are again *study related costs* and in any case would be equal in the randomised groups. The additional cost of PCI is included in the model for patients randomised to PCI.

1.20. Page.8 of 22, line 13..... uses a threshold of £20,000 to £30,000 per QALY gained, with an.....For the purpose of comparing two strategies, the authors should consider the placebo group as a true medical group. In this study, the placebo group underwent intervention and its costs should be noted. Otherwise, this study can be considered a “virtual study” of cost effectiveness.

We accept that there are costs incurred in treating both groups. However, this is why the primary outcome of the analysis is an **incremental** cost-effectiveness ratio which reflect the **difference** in costs between the two treatments. This is a conventional approach in cost-effectiveness analysis.

Probabilistic sensitivity analysis:

1.21. Page.8 of 22, line 42... We did not vary the dosages as we felt this..... This reviewer considers the ORBITA Trial a small study with 195 patients and with a very feasible follow-up. Estimating medications because of the variability of medical prescriptions does not convince this reviewer. The

impression that remains is that the authors do not have this data and are making estimates of medications administered. This is another sign of work fragility.

We respectfully point out that this is not correct. The original ORBITA study recorded detail of all prescriptions for medical therapy at an individual patient level at each study visit. However, for the purpose of the probabilistic sensitivity analysis we sampled from a prior distribution of the average number of patients receiving a specific antianginal medication. As clearly stated in the manuscript we did not apply variation to dosages of individual medications because of the risk of creating scenarios of drug doses that were clinically not feasible. Far from being fragile, this is robust and commonly used methodology for this kind of analysis.

1.22. Page.8 of 22, line 44..... For the costs of the PCI procedure..... The authors report that they considered patients with less comorbidity and with lower health care costs. This does not make sense, because the study aims to analyze cost effectiveness between PCI versus placebo. This is not a clinical study.

We have used the costs for PCI based on NHS England tariffs that were considered most relevant to the patients included in the study i.e. low complexity with few comorbidities. The methodology by which the NHS reaches these tariffs is well beyond the scope of this paper but adequate references have been provided.

1.23. Page.8 of 22, line 44..... The ORBITA patients were not complex..... Again, even though the patients were complex, this complexity would be present in both groups. This argument is not supported.

See response to Comment 1.22 above.

1.24. Page.8 of 22, line 50..... We took 5,000 random samples from the distributions for each relevant parameter. This calculation does not include major cardiovascular events such as AMI, Stroke, Additional intervention and death. This must be considered.

See response to Comment 1.8 above. These events are controlled for by randomisation.

Scenario analysis

1.25. Page.8 of 22, line 58..... refractory symptoms requiring PCI..... In this case, the authors must calculate the exact number of patients. Given the small number of patients the group studied, presumed analysis of patient data is not acceptable.

This is the purpose of the scenario analysis. As described clearly in the methods we explored what proportion of patients treated medically, in increments of 20%, would need to return with refractory symptoms requiring PCI before the incremental cost-effectiveness ratio approached the accepted NHS threshold for cost-effectiveness (GBP 30 000). This how the figure of 80% was reached.

1.26. Page.9 of 22, line 8..... would require less anti-anginal therapy than control patients..... Again. This study must have, in the database, the exact quantity of drugs received.

Again, this is an exploratory scenario analysis the purpose of which was to test assumptions regarding the use of antianginal medication beyond the data derived from the study in order to assess the robustness of our primary outcome.

DISCUSSION

1.27. Page.11 of 22, line 7..... from the only double blind, randomised trial..... Here, the authors must define what is double blind. The operators were not blinded. If the placebo group operators knew that the procedure was fake then it is not double. Maybe just the patients.

The study was double-blind and the effectiveness of masking of both patients and trial clinicians was evaluated. See detailed response to Comment 1.5.

A clear understanding of the study methods is required, but briefly, the randomisation process and randomly allocated cardiac catheterisation procedures (PCI or placebo) were performed by clinical teams in isolation who did not disclose allocation to the clinical teams responsible for all other aspects of the conduct of the trial and clinical care of the patients for the duration of the study. Substantial efforts were made to ensure that patients were also blind to the randomised allocation (sedation, auditory and visual masking) as detailed in the original study methods. Blinding of clinicians and patients was formally evaluated and shown to be highly effective within conventionally accepted bounds.

1.28. Page.11 of 22, line 57..... specific concern is that, over a longer horizon,.... In this case, the authors could find another population of patients with CAD. It is known that the disease is progressive and at the end of 5 years these patients could present with multivessel dysesease.

See detailed response to Comment 1.2 above.

1.29. Page.12 of 22, line 3..... We found that more than 80% of patients would need to return for PCI within 12 months,..... Sorry, this is creative statistics.

We have reported the *calculated* outcome of the scenario analysis described in response to Comment 1.25 above.

1.30. Page.12 of 22, line 6..... in the COURAGE trial over 4.6 years.....The authors cannot consider similarity in the analysis of the two trials. Since the objectives were different and the patients were multivessel. Even so, considering a 12-month follow-up, COURAGE found a crossover occurrence close to 6% per year in the medical group and close to 4% in the PCI group. This in multivessel disease patients.

We accept that the patient populations are quite different, but the COURAGE study remains one of the strongest reference points for our findings. As the reviewer points out, even in the open-label COURAGE trial, which included more complex patients, the rate of cross over from medical therapy to PCI was much less than our scenario analysis suggests would be needed before routine PCI would approach the cost-effectiveness threshold. If anything, this strengthens the conclusions that can be drawn from this exploratory part of our analysis

1.31. Page.12 of 22, line 11..... In the ORBITA trial itself, all patients had already been referred for clinical PCI,..... Do the authors consider that all patients in the placebo group went for percutaneous treatment? If so, it invalidates the study.

See detailed response to Comment 1.10 above.

References

1-Cost-effectiveness of percutaneous coronary intervention versus bypass surgery from a Dutch perspective. Heart. 2015 Dec;101 (24):1980-8.

2-Cost-Effectiveness of Percutaneous Coronary Intervention in Optimally Treated Stable Coronary Patients. August 2008 Circulation Cardiovascular Quality and Outcomes 1(1):12-20 DOI: 10.1161/CIRCOUTCOMES.108.798462.

3-Comparative cost-effectiveness of surgery, angioplasty, or medical therapy in patients with multivessel coronary artery disease. Cost Effectiveness and Resource Allocation volume 16, Article number: 55 (2018)

Reviewer: 2

Reviewer Name

Steffan F. Stella

Institution and Country

McMaster University - Canada

Please state any competing interests or state 'None declared':

None declared

Please leave your comments for the authors below

Thanks for giving me the opportunity to review this manuscript.

Based on clinical results from the ORBITA trial, the authors performed an economic evaluation on percutaneous coronary intervention compared to placebo for stable angina patients from the UK NHS perspective with the intent of providing evidence to inform policy makers and practice.

Major comments:

The authors implied they are using primary patient level data from ORBITA trial. The ORBITA trial reported no difference ($p=0.994$) in the EQ-5D-5L health related quality of life (QoL), with incremental in QoL of 0.03 in each group from baseline to follow up (6 weeks).

The authors presented weighted utility values calculated from the trial data, however it is confusing why they aggregated the utility values for the patients in the baseline as the benefit should be assessed in each group in separate.

Response 1. The reason for using the aggregated utility value at baseline is to estimate the mean utility in all patients with coronary artery disease (CAD), the CAD health state, rather than in each group. The following health states were essentially post-treatment health states of "PCI" or "Medical therapy".

Even correcting that, the lack of difference in effectiveness between groups persists. In this case, given the authors propose a cost effectiveness study, the denominator in the incremental cost effectiveness ratio (ICER) equation will approach to zero, which would make this type of analysis not viable or not necessary.

Response 2. We chose a cost-effectiveness study to demonstrate the extra cost and extra health benefit we would expect to see from a decision to use percutaneous coronary intervention as compared to medical management in this patient group. This is an important question, especially where the intervention is so frequently used and accounts for a good proportion of total health services spending. If extra health benefits are indeed 'small per unit of cost incurred', then this provides important evidence that can be used to improve the efficiency of health services.

A more pragmatic approach would take in consideration that no significant benefit in terms of QALY gained would make one to choose the less costly strategy.

Response 3. Cost-effectiveness conclusions are not informed by arbitrary measures of statistical significance. Rather we seek the probability that choosing one treatment over another will improve

efficiency under conditions of scarce resources, given prior uncertainty in all the parameters that inform that decision. Claxton made these argument in 1999 [2] and it represents a basic principle of cost-effectiveness work today.

In the bottom-line, cost effectiveness studies are justified when one is trying to show if a given benefit in health worth the additional cost allocated on a given strategy.

Response 4. We agree, and make the same point in Response 2

Although it was previously published, the study population should be explored in the manuscript. The reader shouldn't have to refer to the original study to find out the population being studied.

Response 5. The background section has been updated to include further details of the ORBITA trial.

The choice of a Markov model in this context is confusing. Firstly, in the manuscript, PCI and placebo are defined as health states when in fact they are decision branches in the Markov tree.

Response 6. Markov models do not have decision branches, rather they include health states that patients can transition through over time for the purpose of summarising costs and health outcomes. We suggest having a "treatment" or "not" are reasonable health states for this purpose. We wanted to explore the longer-term effects of the two treatments, a Markov model with "treatment" health states enabled us to do this. Markov models are good for modelling situations where people remain in the same state for multiple cycles.

Secondly, no cycle length is defined and the 12 months' extrapolation in the study horizon is problematic.

Response 7. The cycle length was one week. This was inadvertently omitted from the manuscript and has been updated.

It is very hard to extrapolated results from 6 weeks up to 12 months. This study addresses a very specific question and population, and this assumption is completely unreliable.

See response to Reviewer 1, Comment 1.2

Moreover, the Markov model was also used to perform the probabilistic sensitive analysis, although other methods for PSA, like bootstrapping, could be more adequate given patient level information is available for the authors.

Response 8. The probabilistic sensitivity analysis (PSA) approach we used is called Monte Carlo simulation. The goal was to estimate the distribution of the quantities we observed in the primary data by simulating the process that generated it. Bootstrapping is a way of asking, "What would I get if we generated the sample again?" It is an application of Monte Carlo simulation where the user is estimating the distribution of a sample statistic. We feel confident that our approach is suitable.

Authors need to clarify which information they want to provide. If the intent is to show cost differences between strategies, a dedicated cost study would be more appropriate. This would include a more comprehensive look into the costs related to medical therapy and PCI procedure. Ideally, averaging the costs related to each patient would provide the cost dispersion values and parameters across the sample. Additionally, one-way sensitive analysis and Tornado diagram would be of key importance to understand which are the most important cost predictors.

Response 9. This is a cost-effectiveness study and we have justified the reasons for undertaking it, as opposed to a costs-only analysis, see Response 2. As described in the paper, the key costs are related to the PCI procedure. A more detailed costing study would therefore be unlikely to aid decisions about resource allocation for invasive procedures in this patient group. We undertook a “scenario analysis” examining the effects of reduced pharmaceutical cost in the PCI group. We understand the purpose of Tornado diagrams and one-way sensitivity analyses, but suggest these would add no useful information over and above the comprehensive assessment of uncertainty we include in the manuscript. We could potentially do an expected value of information analysis for each model parameter, but given the low chance that reducing variance in any one parameter will change the decision making conclusion, this would be a low priority. While we do agree that such costing analysis could provide greater insights about where to reinvest resources, if disinvestment from PCI for these patients occurred, this is outside the scope of the current analysis.

1. Bang H, Ni L, Davis CE. Assessment of blinding in clinical trials. *Control Clin Trials*. 2004;25(2):143-156. doi:10.1016/j.cct.2003.10.016
2. Claxton K. The irrelevance of inference: a decision-making approach to the stochastic evaluation of health care technologies. *Journal of Health Economics*. 1999;18:341-64.

VERSION 2 – REVIEW

REVIEWER	Hueb, Whady Instituto do Coracao (InCor), Hospital das Clinicas HCFMUSP, Faculdade de Medicina, Universidade de São Paulo, SP, BR
REVIEW RETURNED	01-Oct-2020
GENERAL COMMENTS	The authors have made substantial changes to the text. It certainly gave more consistency to the results. This reviewer will not apply any obstacles to publication.